# Women's decisions regarding family planning use and its determinants in Ethiopia: A systematic review and meta-analysis protocol

**Etsay Woldu Anbesu** [1]*, **Setognal Birara Aychiluhm** [1], **Mussie Alemayehu** [2]

**1** Department of Public Health, College of Medical and Health Sciences, Samara University, Samara, Ethiopia, **2** School of Public Health, College of Health Science, Mekelle University, Tigray, Ethiopia

* etsaywold@gmail.com

**Data Availability Statement:** All relevant data are within the manuscript and its Supporting Information files.

## Abstract

### Background

Low use of contraceptives has many consequences. Despite this effect, less emphasis is given to women's decision-making on family planning use in Ethiopia. Although there are studies conducted in different parts of the country on women's decision-making regarding family planning use, there are inconsistent findings and a lack of national representative data. Thus, this systematic review and meta-analysis aimed to determine the pooled prevalence of women's decision-making regarding family planning use and its determinants in Ethiopia.

### Methods

Preferred Reporting Items for Systematic Reviews and Meta-Analyses guidelines will be followed to develop the review protocol. All observational studies will be retrieved using Medical Subject Heading (MeSH) terms or keywords from the online databases PubMed, CINAHL, Google Scholar, African Journal online, and gray literature. The quality of the studies will be critically assessed using the Joanna Briggs Institute checklist. Heterogeneity among studies will be examined using I-squared statistics. Funnel plots and Egger's test will be used to examine publication bias. The meta-analysis will be performed using STATA version 14 software. Statistical significance will be determined at 95% CI.

### Discussion

Improving women's autonomy in decision-making on reproductive health services, including contraceptive use, has a substantial advantage. There are studies on women's decision-making in family planning use; however, there are inconsistent findings. Therefore, this review protocol aims to determine the pooled prevalence of women's decision-making regarding family planning use and its determinants in Ethiopia. The findings from this systematic review and meta-analysis will help inform policy makers to develop appropriate interventions to improve women's decision making regarding family planning use.

**Funding:** The author(s) received no specific funding for this work.

**Competing interests:** The authors have declared that no competing interests exist.

## Introduction

Women's decision-making power in family planning use is the ability of women to decide freely independently or argue with their husbands or partners about family planning needs and choices [1]. Family planning (FP) is an effort made by couples to limit or space the number of children using family planning methods [2]. Women's decision-making on family planning use is associated with delayed marriage, access to accurate information, free discussions of family planning needs and choices with partners, members of the household, and the community, and independent decisions on fertility regulation, including increased health-seeking behavior to contraceptives [3, 4].

Increased women's decision making regarding family planning provides benefits such as safeguarding the health and rights of women, reducing maternal and child mortality, avoiding unplanned pregnancy and induced abortion, reducing long-term fertility rates, and improving households' economic status [4–7]. Women's decisions on family planning use have numerous benefits for the family, in particular, and the community at large, including the ability to challenge and control at the interpersonal level, increase to make choices on family planning methods, challenge the existing norms and culture to effectively increase their well-being, increase awareness of women's on the types of family planning methods and build their capacity to challenge [8–11]. Women in low-income countries have been deprived of their reproductive health rights [6, 12]. The peculiar factors that predispose women in Africa to deprivation of reproductive rights and their decision-making power during family planning can be social factors such as the low educational status of women and their partners; low level of a couple's communication on family planning methods, young age; area of residency; poor knowledge; lack of information about family planning; economic factors including household wealth status; media exposure; affordable, accessible, and acceptable quality of sexual and reproductive health care services and cultural factors such as attitude towards family planning methods; and social influence by husbands and the community [13–16]. Contraceptive use was much lower due to the challenge of social, cultural, or economic factors, although women in developing countries, particularly in Africa, would like to delay or stop childbearing [4, 17, 18].

In developing countries, contraceptive use is low (40%), and approximately 225 million women have an unmet need for family planning methods [19]. In sub-Saharan Africa, contraceptive use varies from 6.7% in Chad and 72% in Namibia [3]. In Ethiopia, contraceptive use was low (41.4%), and 22% of women had an unmet need for family planning methods. This leads to a high total fertility rate (TFR) of 4.6 children per woman and contributes to maternal mortality 412 per 100,000 live births, neonatal mortality 30, infant mortality 43, and underfive mortality 55 per 1000 live births [20, 21].

Low utilization of contraceptive methods has many consequences, such as high common childhood illness, lack of appropriate health, poor maternal and child health care, increased maternal and child mortality, increased workload of mothers, poor child growth, and unfavorable impact on economic status and growth [2, 22–24]. Factors affecting women's decision-making regarding family planning use were educational level, socioeconomic status, domestic decision-making position, male partner influence, lack of knowledge, gender-based inequalities, and low access to reproductive health services [22, 25–30]. Due to harmful, biased social norms and practices and a lack of financial resources, women are regularly incapable of accessing sexual and reproductive health services. Moreover, lack of access to services reduced women's ability to make choices about their sexual and reproductive health [14].

In Ethiopia, although the national 20-year health sector transformation plan [12] and national guidelines for family planning services [2] emphasize women's decision-making on family planning use, family planning use is still low, and there is a lack of nationally

representative data on women's decision-making on family planning use [20, 21]. There are studies conducted in different parts of the country on women's decision-making in family planning use [22, 25, 31–35]; however, there are inconsistent findings, with a prevalence of women's decisions regarding family planning use ranging from 35.9% to 98%. Moreover, inconsistent findings on factors affecting women's decisions regarding family planning use were reported among the studies, including residence, knowledge, attitude, educational status, occupational status, age, income, husband influence, and number of living children. Women's decision making regarding family planning will be considered whether women have decision-making power regarding delayed marriage, access to accurate information, free discussion about family planning needs and choices with partners, independent decisions on fertility regulation and increased health-seeking behavior to contraceptives in the primary studies. Thus, this systematic review and meta-analysis protocol aimed to determine the pooled prevalence of women's decision-making regarding family planning use and its determinants in Ethiopia.

### Research question

- What is the pooled prevalence of women's decisions regarding family planning use in Ethiopia?

- What are the determinants of women's decisions regarding family planning use in Ethiopia?

### Objectives

- To determine the pooled prevalence of women's decisions regarding family planning use in Ethiopia.

- To identify determinants of women's decisions regarding family planning use in Ethiopia.

## Methods

### Study protocol and reporting

The Preferred Reporting Items for Systematic Review and Meta-analyses (PRISMA) guidelines [36] will be used to prepare the systematic review protocol. The PRISMA-P 2015 checklist will be used for the review report [37] (S1 File).

### Eligibility criteria

All observational studies, including cross-sectional, case–control, cohort and gray literature, performed in Ethiopia will be included. Case reports, case series, and preprints will be excluded from the review; if studies address both quantitative and qualitative findings on women's decision making in family planning use, we will only consider the quantitative results. Studies published in English alone will be included. There will not be a restriction on the publication date.

### CoCoPop/PEO search guide

**Condition.**   Women's decisions regarding family planning use Context; Ethiopia.
**Population.**   Women of reproductive age group (15–49 years).
**Exposure.**   Exposure is a determinant that increases or decreases the likelihood of women's decision-making in family planning use among reproductive-age women in Ethiopia. The determinants can be the educational status of partners, domestic decision-making position,

male partner influence, lack of knowledge on contraceptives, gender-based inequalities, and access to reproductive health services.

**Outcome.** The primary outcome of the study will be the pooled prevalence of women's decision making regarding family planning use. Women's decision making regarding family planning was considered whether women have decision-making power regarding delayed birth spacing, access to accurate information, free discussion about family planning needs and choices with partners, independent decisions on fertility regulation and increased health-seeking behavior to contraceptives in the primary studies. The secondary outcome of the study will be to identify factors affecting women's decision making regarding family plaining use. The selection of independent variables was based on how frequently the variables were reported in the primary studies, and factors that reported more than one study and had consistent classification were included.

## Search strategy and study selection

PubMed, Google Scholar, African Journal Online, CINAHL and gray literature will be used to search studies. In addition, cross-references searching of related studies will be performed from the included studies. An independent search will be performed by two authors. The retrieved studies will be exported to Endnote version 8 reference manager to collect, organize and manage search results [38], and the studies will be screened independently by removing duplicate and irrelevant titles and abstracts. Full text selected studies will be evaluated further for quality. The process of article selection and report results will be reported using the PRISMA chart (*S2 File*). The Medical Subject Heading (MeSH) terms and entry terms will be used to search articles from databases. Then, Boolean operators (OR, AND) will be used to search studies from the online database (S3 File).

## Quality assessments

The quality of studies will be assessed using their title, abstract, and full-text review before the inclusion of studies in the final systematic review and meta-analysis. The Joanna Briggs Institute Meta-Analysis of Statistics Assessment and Review Instrument (JBI-MAStARI) will be used to assess the quality of the studies [39]. The tool emphasizes explicit inclusion and exclusion criteria, standard measurement criteria, study subjects and setting, strategies to deal with confounding, exposure measurement validly and reliably, outcome measurement, and appropriate statistical analysis (S4 File). The full-text articles will be assessed by two authors (EW and SB). A quality scale of 50% and above will be considered and included in the review. Disagreements among reviewers will be discussed by the third author (MA) to reach an agreement.

## Data extraction and management

Independent piloting of the data extraction will be carried out in Microsoft Excel (2016) before the beginning of the actual data extraction. The data extraction tool will contain information on the first author name, publication year, study setting, study design, sample size, prevalence, odds ratio, lower and upper bounds of the confidence interval, log transformation, and standard error of logarithm. The actual data extraction will be performed by the two authors (EW and ZH). Discrepancies in data extraction will be resolved by discussion with a third author (MA). In case of missing data or incomplete reports, the corresponding authors of the studies will be contacted.

## Data synthesis and analysis

Importing of extracted data to STATA version 14 for analysis will be performed. Tukey's test. Freeman Tukey's test will be used for square root transformation of data to avoid variance variability [40].

A random-effects model will be used to determine the pooled prevalence of women's decision-making in family planning use in Ethiopia [41]. Forest plots will be used to present the pooled prevalence and determine women's decision-making in family planning use at a p value of less than 0.05. 05 [42]. Cochran's Q [43] and $I^2$ statistics [44] will be used to identify heterogeneity across studies. The $I^2$ statistic estimates the percentage of variation across studies. $I^2$ values of 25%, 50%, and 75% are representative of low, moderate, and high heterogeneity, respectively. The sources of heterogeneity will be determined using subgroup analysis and meta-regression. Moreover, sensitivity analysis will be performed to investigate the effect size of a single study. A funnel plot will be used to check publication bias using visual observation [42], and an asymmetry of funnel plot indicates publication bias. Statistical Egger's and Begg's tests [45] will be performed to check publication bias, and a p value of $< 0.05$ will be used to declare publication bias.

## Discussion

This systematic review and meta-analysis protocol aims to synthesize the pooled prevalence of women's decisions regarding family planning use and its determinants in Ethiopia. Improving women's autonomy in decision-making on reproductive health services, including contraceptive use, has substantial advantages, including fertility regulation, reduction of child mortality, and improvement of households' child feeding practices [5, 6, 19, 34, 46–48].

Studies have shown that low contraceptive use leads to a lack of proper care for children from household members [24]. Worldwide, approximately six million children die before reaching their first year birthday, and approximately 35 women die every hour due to birth-related complications [49–51]. In developing countries, particularly sub-Saharan African countries, contraceptive utilization was low [23, 24]. In Ethiopia, contraceptive use was low, and there was a higher unmet need for family planning, total fertility rate, and neonatal, infant and maternal mortality [20, 21].

The findings from this systematic review and meta-analysis will help to identify the pooled prevalence and determinants of women's decisions regarding family planning use and help to inform policy makers to develop appropriate interventions to improve women's decisions regarding family planning use. This review protocol may have limitations, including heterogeneity due to differences in study designs, sample sizes, and publication biases. Articles published only in English will be included. Only observational study designs will be included, and interventional and quasiinterventional studies will be excluded from the review.

## Supporting information

**S1 File. PRISMA-P 2015 checklist.**
(DOC)

**S2 File. Diagrammatic presentation of the study selection process for systematic review.**
(DOCX)

**S3 File. Database search terms.**
(DOCX)

**S4 File. JBI critical appraisals for observational studies link:** [https://jbi.global/critical-appraisal-tools](https://jbi.global/critical-appraisal-tools).
(DOCX)

## Acknowledgments

We would like to thank Samara University for the internet and HINARY database website access.

## Author Contributions

**Conceptualization:** Etsay Woldu Anbesu.

**Investigation:** Etsay Woldu Anbesu.

**Methodology:** Etsay Woldu Anbesu, Setognal Birara Aychiluhm, Mussie Alemayehu.

**Writing – original draft:** Etsay Woldu Anbesu.

**Writing – review & editing:** Etsay Woldu Anbesu, Setognal Birara Aychiluhm, Mussie Alemayehu.

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
