## [Decision Letter · Decision Letter 0]

23 Aug 2022

PONE-D-22-02268Women’s decisions regarding family planning use and its determinants in Ethiopia: a systematic review and meta-analysis protocolPLOS ONE

Dear Dr. Anbesu,

Thank you for submitting your manuscript to PLOS ONE. After careful consideration, we feel that it has merit but does not fully meet PLOS ONE’s publication criteria as it currently stands. Therefore, we invite you to submit a revised version of the manuscript that addresses the points raised during the review process.

ACADEMIC EDITOR:

Please, see the additional comments provided below.

We look forward to receiving your revised manuscript.

Kind regards,

Kazeem Babatunde Yusuff

Academic Editor

PLOS ONE

Journal Requirements:

Additional Editor Comments (if provided):

I have carefully reviewed the manuscript titled “Women’s decisions regarding family planning use and its determinants in Ethiopia: systematic review and meta-analysis protocol” and my comments are stated below:

General comment: This proposed study will address an important research question concerning family planning and women’s ability to make decision regarding this within the Ethiopian context. Hence, the study is relevant and warranted especially given challenges that women face in developing settings regarding reproductive health-related choices / decisions. However, I have identified a number of crucial gaps that will require adequate revision before the manuscript is ready for publication. The areas requiring revision are stated below. In addition, the manuscript will generally require professional language editing to improve grammar, syntax, clarity and sentence construction.

Abstract: Appeared relatively well written and focused on the stated aim. The proposed method and brief discussion of the expected results appeared appropriate for the stated aim.

Introduction: Appeared relatively well written but require professional English language editing. Furthermore, authors should address the following points during the revision of the manuscript:

• The review of the existing published body of knowledge in the research area seems inadequate and speculative in some areas. The depth of literature search and summarized discussion of the state of the art of published knowledge in the research area appeared scanty.

• Authors did not provide details of the specific inconsistencies contained in the published findings related to women’s decision-making about family planning and the use of contraceptives in Ethiopia.

• Line 75-77: Lack of details regarding how having greater decision-making power by women will result in the stated benefits.

• Line 77: Lack of details regarding the peculiar factors (social, cultural or economic) that predispose women in Africa to deprivation of reproductive rights? How does this affect their decision-making power during family planning discussions or choices?

• Line 78: Why is that? What factors underline this? Is this cultural or economic? What factors underline this? Is this linked to the lack of decision-making power for women concerning family planning?

• Line 89-92: How? Provide details of how these factors have affected women decision-making power regarding family planning.

• Line 97-99: What were the major findings of these studies? What are the inconsistencies in the data / findings generated from these studies? This is a crucial gap in the introduction section which will provide a solid justification for conducting your SR/MA.

• Line 110: Lack of clarity regarding the specific component / aspect of women's decision-making that will be focused of the SR/MA.

Methods and Discussion: Appropriate for the stated study objective. However, professional language editing is required for grammar, syntax and clarity.

Reviewers' comments:

Reviewer's Responses to Questions

**Comments to the Author**

1. Does the manuscript provide a valid rationale for the proposed study, with clearly identified and justified research questions?

Reviewer #1: No

2. Is the protocol technically sound and planned in a manner that will lead to a meaningful outcome and allow testing the stated hypotheses?

Reviewer #1: No

3. Is the methodology feasible and described in sufficient detail to allow the work to be replicable?

Reviewer #1: No

4. Have the authors described where all data underlying the findings will be made available when the study is complete?

Reviewer #1: No

5. Is the manuscript presented in an intelligible fashion and written in standard English?

Reviewer #1: No

6. Review Comments to the Author

You may also provide optional suggestions and comments to authors that they might find helpful in planning their study.

Reviewer #1: The manuscript is poorly presented and difficult to read and understand. Scientific language is lacking for terms used, research questions and objectives poorly explained, the methodology is difficult to understand and the discussion is lacking depth and coherence.

7. PLOS authors have the option to publish the peer review history of their article (what does this mean?). If published, this will include your full peer review and any attached files.

Reviewer #1: No

---

## [Author Response · Author response to Decision Letter 0]

25 Aug 2022

Dear, editor, and reviewer

Editor Comments 

≠ Authors did not provide details of the specific inconsistencies contained in the published findings related to women’s decision-making about family planning and the use of contraceptives in Ethiopia.

Addressed line 107-113

≠ Line 75-77: Lack of details regarding how having greater decision-making power by women will result in the stated benefits.

Addressed line 73-76

≠ Line 77: Lack of details regarding the peculiar factors (social, cultural or economic) that predispose women in Africa to deprivation of reproductive rights? How does this affect their decision-making power during family planning discussions or choices?

Addressed line 77-84 

≠Line 78: Why is that? What factors underline this? Is this cultural or economic? What factors underline this? Is this linked to the lack of decision-making power for women concerning family planning?

Addressed line 77-84 

≠Line 89-92: How? Provide details of how these factors have affected women decision-making power regarding family planning.

Addressed line 99-102

≠ Line 97-99: What were the major findings of these studies? What are the inconsistencies in the data / findings generated from these studies? This is a crucial gap in the introduction section which will provide a solid justification for conducting your SR/MA.

Addressed line 109-113

≠ Line 110: Lack of clarity regarding the specific component / aspect of women's decision-making that will be focused of the SR/MA.

Addressed line 113-117

Reviewer comments

Reviewer #1: The manuscript is poorly presented and difficult to read and understand. Scientific language is lacking for terms used, research questions and objectives are poorly explained, the methodology is difficult to understand and the discussion is lacking depth and coherence.

Language edited. Revised Manuscript with Track Changes and the clean manuscript was uploaded on the online submission.

---

## [Editor Report · Decision Letter 1]

6 Sep 2022

PONE-D-22-02268R1Women’s decisions regarding family planning use and its determinants in Ethiopia:a systematic review and meta-analysis protocolPLOS ONE

Dear Dr. Anbesu

Thank you for submitting your manuscript to PLOS ONE. After careful consideration, we feel that it has merit but does not fully meet PLOS ONE’s publication criteria as it currently stands. Therefore, we invite you to submit a revised version of the manuscript that addresses the points raised during the review process.

Please, see as stated below

We look forward to receiving your revised manuscript.

Kind regards,

Kazeem Babatunde Yusuff

Academic Editor

PLOS ONE

Journal Requirements:

1. Please note that the Study Protocol article type is only suitable for proposals of studies that have not yet generated results. For further information, please see https://journals.plos.org/plosone/s/submission-guidelines#loc-study-protocols

Please update your Cover Letter to indicate whether you wish to have the manuscript considered as a Study Protocol, and confirm that neither recruitment nor data collection have been completed.

Additional Editor Comments:

1. Kudos to the authors for revising the manuscript for grammar, syntax and clarity. However, few minor edits for grammar remain. Please, check the manuscript to identify the areas requiring minor revision for grammar.

2. The response provided by the authors did not address my previous comment regarding ≠ Line 75-77 in the previous manuscript: Lack of details regarding how having greater decision-making power by women will result in the stated benefits. The revision done in line 73-76 of the revised manuscript did not address the comments. Please, revise this section accordingly.

Addressed line 73-76.

3. The response provided by the authors in line 77-84 of the revised manuscript did not address my previous comment regarding ≠ Line 77: Lack of details regarding the peculiar factors (social, cultural or economic) that predispose women in Africa to deprivation of reproductive rights? How does this affect their decision-making power during family planning discussions or choices? Please, revise the manuscript accordingly.
---

## [Author Response · Author response to Decision Letter 1]

8 Sep 2022

Dear, editor, and reviewer

Editor Comments 

≠ Please update your Cover Letter to indicate whether you wish to have the manuscript considered as a Study Protocol, and confirm that neither recruitment nor data collection have been completed.

Addressed and uploaded on the online submission “Cover letter” file 

Additional Editor Comments

≠ 1. Kudos to the authors for revising the manuscript for grammar, syntax and clarity. However, few minor edits for grammar remain. Please, check the manuscript to identify the areas requiring minor revision for grammar

Checked and uploaded on the “revised manuscript with track changes” file 

≠ 2. The response provided by the authors did not address my previous comment regarding ≠ Line 75-77 in the previous manuscript: Lack of details regarding how having greater decision-making power by women will result in the stated benefits. The revision done in line 73-76 of the revised manuscript did not address the comments. Please, revise this section accordingly.

Addressed line 73-76.

Addressed on line 73-78

≠3. The response provided by the authors in line 77-84 of the revised manuscript did not address my previous comment regarding ≠ Line 77: Lack of details regarding the peculiar factors (social, cultural or economic) that predispose women in Africa to deprivation of reproductive rights? How does this affect their decision-making power during family planning discussions or choices? Please, revise the manuscript accordingly.

Addressed on line 79-88

Reviewer comments

#1: � The link "View Attachments" does not appear

#2: While revising your submission, please upload your figure files to the Preflight Analysis and Conversion Engine (PACE) digital diagnostic tool, https://pacev2.apexcovantage.com/. PACE helps ensure that figures meet PLOS requirements. To use PACE, you must first register as a user. Registration is free. Then, login and navigate to the UPLOAD tab, where you will find detailed instructions on how to use the tool. If you encounter any issues or have any questions when using PACE, please email PLOS at figures@plos.org. Please note that Supporting Information files do not need this step.

As this is a protocol, there is no figure. Only supporting information’s are available and uploaded on the online submission.

---

## [Editor Report · Decision Letter 2]

14 Sep 2022

PONE-D-22-02268R2Women’s decisions regarding family planning use and its determinants in Ethiopia:a systematic review and meta-analysis protocolPLOS ONE

Dear Dr. Anbesu,

Thank you for submitting your manuscript to PLOS ONE. After careful consideration, we feel that it has merit but does not fully meet PLOS ONE’s publication criteria as it currently stands. Therefore, we invite you to submit a revised version of the manuscript that addresses the points raised during the review process.

The manuscript is still filled with grammatical and syntax errors in several areas, and these errors must be addressed with professional editing. A few of the examples of errors are stated below:

Line 75: “increase to make choices”

Line 77: “increase awareness of women's subordination”

Line 81: “such as the low educational status of women and partners;;”

Line 83” “antennal care visits;;”

Line 85: “such us attitude of family planning methods;;”

Line 91-92” “In developing countries, contraceptive use was low (40%), and the unmet need for family planning was 225 million people [20].”

Line 93-94: “In Ethiopia, contraceptive use was low (41.4%), and there was a high unmet need for family planning use (22%).”

Line 98: “Low upconsumption‒consumption”

Line 104: “and reproductive health services”

Line 113-118: “Moreover, there are studies conducted in different parts of the country on women's decision-making in family planning [8, 25, 31-35]; however, there are inconsistent findings, with a prevalence of women’s decisions regarding family planning use ranging from 35.9% to 98%, and different factors affecting women’s decisions regarding family planning use were reported among the studies, including residence, knowledge, attitude, educational status, occupational status, age, income, husband influence, and number of living children.”

Line 180-183: “The primary outcome of the study was the pooled prevalence of women’s decision making regarding family planning use. Women’s decision making regarding family planning was considered whether women have decision-making power regarding delayed marriage”

Line 185-186: “The secondary outcome of the study was to identify determinates of women’s decision making regarding family planning use”

Line 192-193: “online databases and gray literature will be used to search studies.”

Line 194-195: “Independent search strategies will be performed by (EW and SB).”

We look forward to receiving your revised manuscript.

Kind regards,

Kazeem Babatunde Yusuff

Academic Editor

PLOS ONE

Journal Requirements:

1. Please note that the Study Protocol article type is only suitable for proposals of studies that have not yet generated results. For further information, please see https://journals.plos.org/plosone/s/submission-guidelines#loc-study-protocols

Please update your Cover Letter to indicate whether you wish to have the manuscript considered as a Study Protocol, and confirm that neither recruitment nor data collection have been completed.

Additional Editor Comments:

The manuscript is still filled with grammatical and syntax errors in several areas, and these errors must be addressed with professional editing. A few of the examples of errors are stated below:

Line 75: “increase to make choices”

Line 77: “increase awareness of women's subordination”

Line 81: “such as the low educational status of women and partners;;”

Line 83” “antennal care visits;;”

Line 85: “such us attitude of family planning methods;;”

Line 91-92” “In developing countries, contraceptive use was low (40%), and the unmet need for family planning was 225 million people [20].”

Line 93-94: “In Ethiopia, contraceptive use was low (41.4%), and there was a high unmet need for family planning use (22%).”

Line 98: “Low upconsumption‒consumption”

Line 104: “and reproductive health services”

Line 113-118: “Moreover, there are studies conducted in different parts of the country on women's decision-making in family planning [8, 25, 31-35]; however, there are inconsistent findings, with a prevalence of women’s decisions regarding family planning use ranging from 35.9% to 98%, and different factors affecting women’s decisions regarding family planning use were reported among the studies, including residence, knowledge, attitude, educational status, occupational status, age, income, husband influence, and number of living children.”

Line 180-183: “The primary outcome of the study was the pooled prevalence of women’s decision making regarding family planning use. Women’s decision making regarding family planning was considered whether women have decision-making power regarding delayed marriage”

Line 185-186: “The secondary outcome of the study was to identify determinates of women’s decision making regarding family planning use”

Line 192-193: “online databases and gray literature will be used to search studies.”

Line 194-195: “Independent search strategies will be performed by (EW and SB).”
---

## [Author Response · Author response to Decision Letter 2]

15 Sep 2022

Dear, editor, and reviewer

Editor Comments 

The manuscript is still filled with grammatical and syntax errors in several areas, and these errors must be addressed with professional editing. A few of the examples of errors are stated below:

Line 75: “increase to make choices”

Corrected line 75-76 

Line 77: “increase awareness of women's subordination”

Corrected line 77

Line 81: “such as the low educational status of women and partners;;”

Corrected line 81

Line 83” “antennal care visits;;”

Corrected line 83

Line 85: “such us attitude of family planning methods;;”

Corrected line 85

Line 91-92” “In developing countries, contraceptive use was low (40%), and the unmet need for family planning was 225 million people [20].”

Corrected line 90-91

Line 93-94: “In Ethiopia, contraceptive use was low (41.4%), and there was a high unmet need for family planning use (22%).”

Corrected line 92-93

Line 98: “Low consumption‒consumption”

Corrected line 97

Line 104: “and reproductive health services”

Corrected line 102-103

Line 113-118: “Moreover, there are studies conducted in different parts of the country on women's decision-making in family planning [8, 25, 31-35]; however, there are inconsistent findings, with a prevalence of women’s decisions regarding family planning use ranging from 35.9% to 98%, and different factors affecting women’s decisions regarding family planning use were reported among the studies, including residence, knowledge, attitude, educational status, occupational status, age, income, husband influence, and number of living children.”

Corrected line 111-117

Line 180-183: “The primary outcome of the study was the pooled prevalence of women’s decision making regarding family planning use. Women’s decision making regarding family planning was considered whether women have decision-making power regarding delayed marriage”

Corrected line 174-176

Line 185-186: “The secondary outcome of the study was to identify determinates of women’s decision making regarding family planning use”

Corrected line 179-180

Line 192-193: “online databases and gray literature will be used to search studies.”

Corrected line 186

Line 194-195: “Independent search strategies will be performed by (EW and SB).”

Corrected line 188

---

## [Editor Report · Decision Letter 3]

29 Sep 2022

Women’s decisions regarding family planning use and its determinants in Ethiopia:a systematic review and meta-analysis protocol

PONE-D-22-02268R3

Dear Dr. Anbesu,

We’re pleased to inform you that your manuscript has been judged scientifically suitable for publication and will be formally accepted for publication once it meets all outstanding technical requirements.

Kind regards,

Kazeem Babatunde Yusuff

Academic Editor

PLOS ONE
---

## [Editor Report · Acceptance letter]

3 Oct 2022

PONE-D-22-02268R3 

Women’s decisions regarding family planning use and its determinants in Ethiopia: A systematic review and meta-analysis protocol 

Dear Dr. Anbesu:

I'm pleased to inform you that your manuscript has been deemed suitable for publication in PLOS ONE. Congratulations! Your manuscript is now with our production department. 

Kind regards, 

on behalf of

Dr. Kazeem Babatunde Yusuff 

Academic Editor

PLOS ONE